# Sinonasal Side Effects of Chemotherapy and/or Radiation Therapy for Head and Neck Cancer: A Literature Review

**DOI:** 10.3390/cancers14092324

**Published:** 2022-05-07

**Authors:** Giuseppe Riva, Ester Cravero, Claudia Pizzo, Marco Briguglio, Giuseppe Carlo Iorio, Chiara Cavallin, Oliviero Ostellino, Mario Airoldi, Umberto Ricardi, Giancarlo Pecorari

**Affiliations:** 1Division of Otorhinolaryngology, Department of Surgical Sciences, University of Turin, 10126 Turin, Italy; ester.cravero@unito.it (E.C.); claudia.pizzo@unito.it (C.P.); marco.briguglio@unito.it (M.B.); giancarlo.pecorari@unito.it (G.P.); 2Division of Radiotherapy, Department of Oncology, Città della Salute e della Scienza Hospital of Turin, 10126 Turin, Italy; giorio@cittadellasalute.to.it (G.C.I.); ccavallin@cittadellasalute.to.it (C.C.); 3Division of Medical Oncology 2, Department of Oncology, Città della Salute e della Scienza Hospital of Turin, 10126 Turin, Italy; oostellino@cittadellasalute.to.it (O.O.); mairoldi@cittadellasalute.to.it (M.A.); 4Department of Oncology, University of Turin, 10126 Turin, Italy; umberto.ricardi@unito.it

**Keywords:** head and neck cancer, radiotherapy, rhinitis, rhinosinusitis, chemotherapy, smell, olfactory disorders, mucociliary clearance

## Abstract

**Simple Summary:**

Rhinosinusitis and smell alterations are common side effects during and after radiotherapy and chemotherapy for head and neck cancer. The assessment of sinonasal complaints is important to increase patients’ quality of life. The aim of this review is to summarize and analyze our current knowledge of the sinonasal side effects of chemotherapy and/or radiation therapy for head and neck cancer, with a specific focus on mucosal and olfactory disorders.

**Abstract:**

Radiotherapy and chemotherapy represent important treatment modalities for head and neck cancer. Rhinosinusitis and smell alterations are common side effects in the sinonasal region. This review will summarize and analyze our current knowledge of the sinonasal side effects of chemotherapy and/or radiation therapy for head and neck cancer (HNC), with a specific focus on mucosal and olfactory disorders. A review of the English literature was performed using several databases (PubMed, Embase, Cochrane, Scopus). Fifty-six articles were included in qualitative synthesis: 28 assessed mucosal disorders (rhinitis or rhinosinusitis), 26 evaluated olfactory alterations, and 2 articles addressed both topics. The incidence and severity of olfactory dysfunction and chronic rhinosinusitis were highest at the end of radiotherapy and at three months after treatment and decreased gradually over time. Smell acuity deterioration and chronic rhinosinusitis seemed to be related to radiation dose on olfactory area and nasal cavities, but different degrees of recovery were observed. In conclusion, it is important to establish the severity of chronic rhinosinusitis and olfactory dysfunction in order to find strategies to support patients and improve their quality of life.

## 1. Introduction

Head and neck cancer (HNC) is the sixth most common type of cancer worldwide and consists of a heterogeneous group of malignancies that may determine important morbidity to affected patients [1,2]. Smoking and alcohol consumption represent the major risk factors for the development of squamous cell carcinoma (SCC) in the head and neck district. Human papillomavirus (HPV) infection plays a role as an etiologic factor of oropharyngeal SCC, especially in tonsils and the tongue base [3].

HNC treatment includes surgery, radiotherapy (RT), and chemotherapy (CT), which are employed according to the tumor stage and primary site involved [4]. The management of early-stage cancer is usually a single modality, either surgery or radiotherapy. On the contrary, the locally advanced tumor has a multimodal treatment: either surgery followed by adjuvant radiotherapy or chemo-radiotherapy (CT-RT), or definitive CT-RT. For recurrent and metastatic diseases that are not susceptible to surgical approaches, chemotherapy or immunotherapy is suggested. Finally, in the last decade, electrochemotherapy (ECT) emerged as a curative or palliative treatment for selected cases of recurrent oral and oropharyngeal cancer [5,6].

Surgery remains the main treatment modality for most of HNC. However, radiotherapy provides an important contribution to HNC management. The radiation dose usually ranges from 60 to 70 Gy, mainly depending on adjuvant or definitive initial intent [4]. The risk of long-term toxicity from RT is dose-dependent with organ-specific susceptibility. Indeed, organs at risk (OARs) protection should be adequately planned before starting RT. The introduction of intensity-modulated radiotherapy (IMRT) allowed us to reduce side effects, providing better balance between target coverage and the sparing of adjacent organs (OARs) [7]. In particular, IMRT guaranteed fewer xerostomia and dysphagia [8].

Platinum-based compounds represent the standard radiosensitizer regimen in the treatment of HNC. Chemotherapeutic drugs such as cisplatin, 5-fluorouracil, and docetaxel may be administered as inductive chemotherapy in the selected case of locally advanced HNC or in recurrent/metastatic tumors [4]. Reducing RT and CT side effects is important to avoid treatment breaks that may negatively impact clinical outcomes [9]. For example, platinum-based compound ototoxicity should be taken into consideration during treatments [4].

Smell and taste alterations are common side effects in patients undergoing CT and/or RT for HNC [10,11]. Olfactory and gustatory dysfunction negatively affects appetite driving to an inadequate food intake, and consequent weight loss. Finally, smell and taste changes may cause a deterioration in quality of life [12].

Radiation therapy determines the damages of irradiated mucosa. This can sometimes lead to severe complaints during and immediately after RT for HNC. Moreover, concomitant CT can induce more toxicity at a cumulative dose, due to a cytotoxic effect on rapidly growing non-cancer cells, such as mucosal cells [13]. In the sinonasal district, CT-RT determines the destruction of mucosal cilia of epithelial cells, leading to the impairment of mucociliary clearance, which increases the risk of developing chronic rhinosinusitis [14]. Moreover, an inflammatory infiltrate can be observed at nasal cytology [15].

This literature review will summarize and analyze our current knowledge of the sinonasal side effects of chemotherapy and/or radiation therapy for HNC, with a specific focus on mucosal and smell disorders.

## 2. Materials and Methods

A review of the English literature was performed using several databases (PubMed, Embase, Cochrane, Scopus, accessed on 31 December 2021) in order to identify articles published before 31 December 2021.

A primary search was performed using the terms “(head and neck cancer) AND (chemotherapy OR radiotherapy OR radiation therapy) AND (nasal OR sinus OR rhinitis OR rhinosinusitis OR smell OR olfactory)”. Search strategies were adapted for each database.

The inclusion criteria were clinical trials, cohort studies, case-control studies, and case series, regarding olfactory dysfunction and chronic rhinosinusitis as side effects of CT and/or RT in HNC patients. Exclusion criteria were as follows: non-human studies, non-English literature, mucosal and olfactory disorders not related to CT and/or RT for HNC.

The abstracts of all suitable articles were examined using the inclusion criteria for applicability. The references of the selected publications were reviewed, in order to identify further reports that were not found by database searching. Two independent reviewers (GR, EC), working separately extracted the data from all the eligible studies, which were subsequently cross-checked. All retrieved full-texts articles were included in the review by a consensus of all the authors. The review included prospective, cross-sectional, and retrospective studies. No studies concerning the sinonasal side effects of target therapy (cetuximab) and immunotherapy were found.

## 3. Results

### 3.1. Literature Review

A total of 5346 published papers were identified using database searches (Figure 1). After abstract screening for eligibility, 71 articles were considered eligible. Among these, we included 56 articles in qualitative synthesis after a full-text assessment. The other 15 papers were excluded because they were systematic reviews (4), they were not in English (4), they did not include HNC patients (4), or because mucosal and/or olfactory disorders were not related to CT and/or RT for HNC (3).

The included studies were published between 1975 and 2021 and conducted in several countries worldwide. Among the 56 selected studies, 28 assessed mucosal disorders (rhinitis or rhinosinusitis), 26 evaluated olfactory alterations, and 2 articles addressed both topics [15,16]. Table 1 highlights the main results concerning sinonasal mucosa disorders, while Table 2 reports publications on olfactory dysfunction.

### 3.2. Sinonasal Mucosa Disorders

Among 30 publications concerning sinonasal mucosa disorders as sides effects of CT and/or RT in HNC patients, six studies were prospective [15,16,20,23,35,38], 10 cross-sectional [17,18,21,24,25,29,30,33,36,43], and 14 retrospective [14,19,22,26,27,28,31,32,34,37,39,40,41,42] (Table 1). Sample sizes ranged from 9 to 1134 patients. Men were predominant in all the publications, except for three papers [35,37,39]. Age ranged from 7 to 84 years. A control group was present in six studies [17,18,20,23,33,36].

The most frequent cancer sites were nasopharynx, nasal cavity, and paranasal sinuses, followed by oropharynx, oral cavity, larynx, hypopharynx, skull base, and parotid gland. Nasopharynx was the only tumor site in all the patients in twenty articles [16,18,19,21,22,23,24,25,26,27,28,30,31,32,33,34,35,38,41,43], whereas one article focused on nasopharynx and nasal cavity [17], one on nasal cavity and paranasal sinuses [40], and another one on patients with nasopharynx and larynx tumors [29]. Moreover, one study included only laryngeal cancer [36], and in one paper, the tumor sites were larynx, oropharynx, oral cavity, and hypopharynx [20]. Tumor stage was reported by 21 articles and ranged from I to IV according to TNM classification [14,15,16,20,22,24,26,28,29,30,31,32,33,34,35,36,37,38,39,41,43].

Treatments included: RT alone in 27 papers [14,16,17,18,19,20,21,22,23,24,25,26,27,28,29,30,31,32,34,35,37,38,39,40,41,42,43], surgery and adjuvant RT in 5 papers [15,20,29,36,42], CT-RT in 20 papers [14,15,16,20,21,22,24,25,26,28,30,31,32,33,34,39,40,41,42,43], and trimodal therapy in 3 papers [15,36,42]. Only fourteen studies described the type of RT that was used. In particular, four studies included two-dimensional radiation therapy (2D-RT) [26,31,33,38], six papers included three-dimensional conformal radiation therapy (3D-CRT) [14,26,30,31,33,39], and fourteen papers included intensity-modulated radiation therapy (IMRT) [14,16,24,26,28,30,31,32,33,34,38,39,41,43]. Two studies evaluated brachytherapy [22,39]. RT dose to nasal cavities was detected only in three studies, and ranged from 13.59 to 56.38 Gy [15,18,43].

The main chemotherapeutic drugs administered in the studies were platinum compounds, pyrimidine compounds, and taxanes. No study evaluated CT alone.

Diagnosis and/or complaints of chronic rhinosinusitis (CRS) were performed using subjective and objective measurements: questionnaires (e.g., Sino-Nasal Outcome Test–SNOT), mucociliary clearance (saccharine test), clinical examination by nasal endoscopy, cultures, nasal biopsy, nasal cytology, computed tomography and magnetic resonance imaging (MRI). No study used only subjective measurements. Thirteen papers included both subjective and objective tools [15,16,17,24,27,30,31,33,35,36,37,38,43].

Nasal endoscopy was performed in 10 studies [15,19,24,26,27,31,33,35,36,43]; two of these used the Lund Endoscopic Staging System, assessing the appearance of nasal endoscopy findings (polyps, oedema, discharge, scarring and crusting), with a total score ranging from 0 to 20 [24,35]. A general clinical examination was included in three articles [14,39,42]. A saccharine test was employed in six studies in order to investigate mucociliary clearance rates [17,19,20,23,29,43]. It consisted of the placement of saccharin on the floor of nasal cavity behind the anterior end of the inferior turbinate. Then, the subjects were asked to swallow every 30 s and to report the first change in taste sensation. The time from the placement of saccharin to the perception of sweetness was noted as the mucociliary clearance time.

The patients underwent a radiological assessment in seventeen studies: computed tomography in six articles [16,19,24,35,37,38], MRI in five articles [22,30,32,34,43], and both computed tomography and MRI in five papers [14,26,27,28,41]. In only one study did the authors use radiographs [21]. The extent of rhinosinusitis was graded using the Lund-Mackay staging system, based on computed tomography findings in six studies [16,19,24,35,37,38], and on MRI findings in three papers [22,32,34]. The Lund-Mackay staging system was based on the opacification of each sinus (maxillary, frontal, anterior ethmoidal, posterior ethmoidal, sphenoidal, and the ostiomeatal complexes), assigning a score between 0 and 2 (0, no abnormality; 1, partial opacification; 2, total opacification) for a total score ranging between 0 and 24. The computed tomography is generally used for rhinosinusitis staging. However, some studies used MRI [22,32,34].

Three studies included nasal cytological examination [15,33,36], and four papers analyzed histopathologic findings based on biopsies [18,23,36,37]. The bacteriology of RT-induced rhinosinusitis, detected by cultures of maxillary sinus specimens, was reported in three studies [21,25,40].

Patients were evaluated before and after RT in fifteen studies [15,16,17,19,20,22,24,28,29,30,32,34,35,41,43]. The minimum time of assessment after treatment was three months [15], and the maximum was 117 months [17]. Fourteen articles evaluated patients only after RT [14,18,21,23,25,26,27,31,33,36,37,39,40,42], with a minimum time of assessment of three months [42] and a maximum of 26–54 years [26].

The percentage of patients with rhinosinusitis after RT ranged from 7% [42] to 86.1% [30]. The most common isolates in the post-RT CRS group were *Staphylococcus aureus*, followed by *Streptocuccus viridans* and *Pseudomonas aeruginosa* [21,25,40].

In four articles, a higher saccharin perception time was found after RT [19,20,23,29]. Kılıç et al. described a higher saccharin perception time in patients receiving RT dose >60 Gy and in patients receiving CT concurrent to RT [29]. On the contrary, Kamel et al. did not find any correlation between RT dose and mucociliary clearance delay time, endoscopic findings, and Lund-Mackay score [19]. Hu et al. found a decreased saccharin transit time after RT [23].

Maxillary sinuses were the most involved, followed by the anterior ethmoid [14,19,22,28,34]. Park et al. described a higher bilateral CRS percentage in non-RT group (85.7% vs. 60%), and, although RT itself was not associated with sinus surgery, concurrent CT was significantly associated with the need for surgery [14]. Advanced T stage (but not RT dose) was positively associated with the incidence of sinus abnormality in the fifth year after RT [22,28]. In the study by Lu et al., patients in the RT-alone and any-RT groups exhibited an increased risk of CRS compared to patients in the no-RT group (hazard ratio: 6.76 and 2.91, respectively) [42].

The incidence of sinusitis peaked at 3–9 months after RT and showed a trend toward stabilization after 1 year [28,34]. Nasal irrigation reduced CRS post-RT in patients with nasopharyngeal carcinoma [24]. Moreover, higher incidence of CRS after RT was observed in patients who used a nasal sprayer instead of a nasal irrigator [30]. Fewer nasal complaints (overall symptoms, blocked nose, and headache), better quality of life, and less severe endoscopic findings were found in the steroid group at 3 and 6 months after RT [35].

Two articles did not find any significant influence of the RT delivery method for any type of complication [31,33]. Choanal stenosis was found in 4.3% to 23% of cases after RT [23,26,31,38] and negatively affected quality of life [31].

Histologically, areas of ciliary loss, intercellular and intracellular vacuolation, and ciliary dysmorphism, and a decreased number of submucosal gland openings and ciliary areas were found after RT [18,23]. Stoddard et al. found a higher percentage of neutrophilic inflammation and squamous or mucous cell metaplasia in the study group without cytological atypia; furthermore, there was no correlation between cytological changes and symptoms, endoscopic findings, age, smoking, and tumor stage [40].

Nasal cytology showed a radiation-induced rhinitis with neutrophils [15,33]. Furthermore, mucous cell metaplasia appeared in patients during RT [15,33,37]. Mucous cell metaplasia was also found in 20% of laryngectomized patients [36]. Increased squamous metaplasia and subepithelial edema and a higher Lund-Mackay score were observed in radiation-induced CRS compared to CRS without nasal polyps [37]. Riva et al. did not find any correlation between cytological changes and symptoms, endoscopic findings (turbinate hypertrophy, mucosal hyperemia, nasal secretions), age, smoking, tumor stage, and adjuvant RT after total laryngectomy [36]. Yin et al. showed that the patients who received IMRT at a dose less than the threshold had the least damaged nasal mucosa morphology, and functional impairment scores were highest 3 months after RT, with a significant relationship between the turbinate thickness ratio and the radiation dose [43].

### 3.3. Olfactory Dysfunction

Among 28 publications concerning olfactory dysfunction as a side effect of CT and/or RT in HNC patients, 16 studies were prospective [15,16,44,46,47,48,49,52,54,55,58,59,60,64,65,67], 11 cross-sectional [50,51,53,56,57,61,62,63,66,68,69], and only one was retrospective [45] (Table 2). Sample size ranged from 10, in two prospective studies [15,65], to 205 patients, in a cross-sectional study [50]. Most patients were men. Only two papers reported a preponderance of women in their sample [57,58]. Age ranged from 11 to 88 years. Seven articles included a control group [46,49,56,57,61,63,69], five were cross-sectional, and two were prospective.

The most frequent cancer sites were nasopharynx, nasal cavity, and paranasal sinuses, followed by oropharynx, oral cavity, larynx, hypopharynx, skull base, and parotid gland. Nasopharynx was the only tumor site in all the patients in six articles [16,46,47,56,57,61], while one article focused on nasopharynx and nasal cavity [66]. Thirteen papers did not include nasopharynx as a tumor site [49,50,51,53,54,58,59,60,62,63,64,65,68,69]. Tumor stage was reported by nineteen articles and ranged from I to IV according to TNM classification [15,16,33,36,46,47,50,51,54,56,58,60,61,62,64,65,66,68,69].

Treatments included: RT alone in three papers [44,45,46], surgery and adjuvant RT in four papers [50,51,52,53], CT-RT in ten papers [16,47,48,49,52,55,56,57,61,65], trimodal therapy in nine papers [15,60,62,63,64,66,67,68,69], and ECT and RT in one paper [58]. Five studies included two-dimensional radiation therapy (2D-RT) [44,45,46,56,57], four studies included three-dimensional conformal radiation therapy (3D-CRT) [48,49,57,66], and five included intensity-modulated radiation therapy (IMRT) [16,57,65,66,68]. The other 14 articles did not specify the type of RT employed. RT dose to the olfactory area was reported in six articles and ranged from <10 to 75 Gy [15,44,45,48,52,55].

The evaluation of CT alone was present in only one paper [59]. The main chemotherapeutic agents administered in the studies were: platinum compounds, pyrimidine compounds, and taxanes. The agent administered for ECT was Bleomycin.

Time of assessment included evaluations before and after RT in fifteen studies [15,16,44,46,47,48,49,52,54,55,58,59,60,64,67], while it was only after treatment in twelve articles [50,51,53,56,57,61,62,63,65,66,68,69]. In one study, patients were evaluated during RT [45]. The longest follow up time was 10 years [69], while the shortest was 2.5 months after the end of RT [60]. The times of assessment in the publication that evaluated only CT were before and immediately after the first, second, and third cycle of therapy [59].

Olfactory function was evaluated by means of psychological tests in eleven papers [44,46,48,49,55,56,59,63,64,65,69], self-report instruments in eight papers [45,50,51,53,54,58,60,62], and through a combination of both tools in nine articles [15,16,47,52,57,61,66,67,68].

The psychological tests measured the three main olfactory abilities concurrently in eight studies [15,46,47,48,57,59,63,64]. They included odor detection threshold (ODT), odor discrimination (OD), and odor identification (OI). ODT was assessed with the use of amyl acetate and eugenol, n-butyl alcohol, or n-butanol. A total score (TDI—Threshold Discirmination Identification) was calculated in the studies that used Sniffin’ sticks for olfactory assessement [15,47,48,57,59,63]. Other objective measurements were olfactory event-related potential testing [61] and olfactory bulb volume in MRI [56]. Olfactory event-related potential testing consisted of a selective stimulation with 40 randomized olfactory stimuli, using phenyl ethyl alcohol and hydrogen sulphide in nitrogen as odorants, presented through a Teflon nasal outlet that was placed into the nasal vestibule [61].

Subjective measurements were obtained through self-report instruments, like Visual Analogue Scale (VAS, 0–100 or 0–10), 6-item Hyposmia Rating Scale, or self-reported smell (EORTC QLQ-H&N35, CCS, SNOT-22, Vanderbilt Head and Neck Symptom Survey version 2.0, AHSP, EQ-5D VAS, MDASI-HN, ASBQ).

Twenty articles reported substantial smell deterioration, with specific differences in odor identification and odor discrimination. ODT significantly decreased during RT, and its baseline levels had still not recovered at different months after the treatment [44,46,47,52,55,56,57,63,67], such as OD in two papers [48,63]. In two cases, ODT did not show variations during and after the radiotherapy treatment [48,64]; similarly, OD did not change in the other two papers [47,57]. OI decreased in three papers [52,56,63], while in two papers, it did not show substantial differences before and after treatment [47,57]. A significant reduction of OI was observed in one study during RT, but showed a partial recovery at 3 months follow-up [67]. On the contrary, in another publication, OI was stable at 2–6 weeks after the beginning of radiotherapy, but showed a decrease after treatment at a long term evaluation [48].

Eight studies did not demonstrate significant differences in olfactory alterations between irradiated and non-irradiated patients [49,50,53,54,58,63,65,69].

Four studies reported an impairment of quality of life [60,61,62,67]. Smell disorders were predictors of depression and anxiety [62], and had a positive correlation with the VAS scale [61].

## 4. Discussion

Head and neck cancer treatment is often challenging and multimodal, including radiation therapy and chemotherapy [4]. In recent decades, the development of new radiotherapy techniques has guaranteed a reduction in OARs complications. In particular, RT technology has developed from 3D-conformal planning to IMRT, and the amount of radiation applied to the surrounding tissue has been minimized [7]. However, since the percentage of long-term HNC survivors has been increasing [1], higher attention should be paid to late side effects.

The aim of this review was to summarize the current knowledge of sinonasal side effects of chemotherapy and/or radiation therapy for HNC, with a specific focus on mucosal and smell disorders. Indeed, these complications are often overlooked by physicians, but may negatively affect patients’ quality of life.

The prevalence of sinonasal mucosa disorders after RT for HNC is very heterogeneous among studies, ranging from 8% to 86.1%. The 5-year incidence of post-irradiation CRS reached 16.7% in the study by Hsin et al. conducted on 102 patients with nasopharyngeal carcinoma (NPC) [32]. However, the applicability of this information is limited because this study investigated only patients with NPC, and not with the HNC of other sites. Furthermore, potential confounders such as CT or previous sinonasal disorders have not been properly considered or controlled. Radiation-induced choanal stenosis is a rare late toxicity, observed in 4.3% to 23% of cases after RT [23,26,31,38]. It leads to serious difficulty in breathing through, and discharge from, the nose. When it comes to improving these symptoms, transnasal endoscopic surgery is often needed [31].

The results obtained by Lu et al. revealed that the risk of CRS was significantly higher in the RT groups compared to the no-RT group. In particular, the 5-year cumulative incidence rates of CRS in the RT-alone, any-RT, and no-RT groups were estimated as 12%, 9.3%, and 4.5%, respectively. The hazard ratios for CRS were 6.76 and 2.91 in the RT-alone and any-RT groups, respectively, compared to the no-RT group. Moreover, the HNC site is another factor associated with the risk of post-treatment CRS: the patients with NPC had the highest incidence of CRS (HR 4.54), while patients with oral cancer had the lowest (HR 0.29) [42].

The incidence of rhinosinusitis peaked at 3–9 months after RT and showed a trend toward stabilization after one year [27,34]. The highest incidence and severity of sinus mucosa disease were found at 3 months after RT (about 67% of cases), which decreased gradually over time. Advanced tumor stage and smoking habit were predisposing factors for sinus mucosa disease, while age, sex, RT dose, and nodal status were not. On the contrary, no factors could predict sinus mucosa disease improvement after RT [22]. In the study by Su et al., a lot of patients with CRS before treatment suffered aggravated symptoms after RT, but 75.3% of patients without CRS before RT developed it after RT. Advanced tumor stage, invasion of the nasal cavity, and nasal irrigation, but not CT or RT dose, were positively associated with the incidence of rhinosinusitis after RT [28]. In the paper by Hamilton et al., nasal crusting, CRS, and epistaxis were quite common (nasal crusting in 16% of patients, epistaxis in 16%, and chronic sinusitis in 8% of cases), attributable to the higher proportion of subjects treated with high dose RT to the nasopharynx in their study [39].

The histopathological features of radiation-induced CRS are different from non-post-RT CRS. Kuhar et al. observed that patients with RT-induced CRS exhibited greater squamous metaplasia and subepithelial edema compared to patients with CRSsNP, and decreased eosinophilia and basement membrane thickening compared to patients with CRSwNP [37]. Riva et al. showed that a radiation-induced rhinitis with neutrophils and sometimes bacteria occurred in 70% of cases and persisted after 1 month; mucous cell metaplasia appeared in 10% of patients during RT and disappeared after 3 months, and squamous cell metaplasia was observed in 10% of cases, only after the end of RT [15]. These results are in agreement with their previous study on the late effects of RT for nasopharyngeal cancer that showed 40% of neutrophilic rhinitis, 20% squamous cell metaplasia, and 13% mucous cell metaplasia, after a median follow-up of 59 months [33]. The most common isolates in the post-RT CRS group were *Staphylococcus aureus*, followed by *Streptocuccus viridans* and *Pseudomonas aeruginosa* [21,25,40].

Few studies have investigated the relationship between RT dose and radiation-induced damage to the nasal mucosa. Indeed, RT dose to nasal cavities was detected only in three studies, and ranged from 13.59 to 56.38 Gy [15,18,43]. Therefore, there is insufficient information on the radiation tolerance of the nasal mucosa. Yin et al. investigated 66 patients and assessed the radiation tolerance of the nasal mucosa by performing the modified saccharin test, endoscopy test, MRI, and a SNOT-20 survey. The results showed that there was a threshold radiation dose, and that above such an RT dose, nasal tissues may not recover from radiation-induced damage. The threshold doses of IMRT ranged from 37 to 40 Gy. A low dose of IMRT (inferior to the threshold dose) was associated with higher mucocilia transport rate, better endoscopy test score, and improved SNOT-20 score [43]. Moreover, an association between turbinate thickness ratio and radiation dose was observed. Riva et al. noticed a significant correlation between mean dose (Dmean) and near maximum dose (D2%) to inferior turbinates and neutrophilic rhinitis, and between D2% to inferior turbinates and mucous cell metaplasia at the end of RT [15].

Treatment for radiation-induced CRS was reported by four studies, and included functional endoscopic sinus surgery [14,23,27,37]. An algorithm for radiation-induced CRS therapy, including medical and surgical options, has never been proposed.

Smell perception plays an important role in life experiences and influences every aspect. Olfactory dysfunction can limit daily life activities and have an adverse effect on nutritional status. Concerning HNC, olfactory disorders may be secondary to direct damage caused by tumors, or to treatments, including surgery, RT, and CT. In particular, RT and CT can alter smell perception by damaging the olfactory epithelium and/or nerves.

The percentage of patients with olfactory impairment after RT ranged from 7% to 76%. Smell disorders appeared during RT and decreased after treatment, remaining higher than non-irradiated patients [45,51,56,57,61,62,63,65,66,68]. Studies that did not find any differences in smell perception between irradiated and non-irradiated patients mainly included subjects affected by non-nasopharyngeal tumors [50,53,69]. Therefore, this lack of difference could be explained by the fact that olfactory epithelium was outside the radiation fields in these patients.

A number of studies have investigated different aspects of olfactory impairment during and after RT. In particular, odor detection threshold (ODT), odor discrimination (OD), and odor identification (OI) have been analyzed. ODT was measured in thirteen studies and worsened during RT [15,33,36,44,46,47,48,52,55,56,59,64,67]. Only three studies did not find ODT alterations after RT, assessing smell at least 12 months after completing the treatment [15,48,64]. OD was measured in seven publications [15,33,36,47,48,59,64]. Among these, four studies did not report OD alterations at a long-term evaluation [15,33,47,48]. Finally, OI was measured in sixteen studies and an alteration was found in 22–63% of patients [15,33,36,46,47,48,49,52,56,59,64,65,67,68,69]. The reason for better OD and OI results compared to ODT may be the supraliminary and centrally integrated nature of OD and OI functions, and thus theoretically outside the field of irradiation. Globally, threshold discrimination identification score (TDI) was lower during RT and totally or partially recovered after treatment [15,57].

Only few studies investigated the role of different RT techniques on smell disorders. Concerning studies that included 2D-RT and 3D-CRT, there was a substantial smell deterioration [44,45,46,48,49,56,57,66], even though Riva et al. did not observe significant long-term differences for subjective hyposmia, ODT, OD, and OI between different radiation techniques for NPC (2D-RT/3D-CRT vs. IMRT) [57]. An impaired olfactory function was also found in five articles that included IMRT [16,57,65,66,68]. Nevertheless, Epstein et al. reported decreased OI in 3 patients (33%) during treatment with smell recovery after RT [65].

A worse olfactory function was observed in patients receiving ‘high dose’ compared to ‘low dose’ to the olfactory epithelium. After therapy, 40% and 7% reported subjective olfactory decline in high and low RT dose groups, respectively [52].

A small part of cited studies was derived from 2019–2022, which coincides with the SARS-CoV-2 pandemic. Since dysfunction of smell and taste could be an effect of SARS-CoV-2 infection, it may partially modulate results in the most recent studies.

Only two studies described both sinonasal mucosal disorders and olfactory disfunction after RT treatment [15,16]. Wang et al. performed patients’ evaluations before and 12 months after RT, observing an increase in sinonasal mucosa and smell disorders after treatment. Moreover, olfactory alterations correlated with total and ethmoid Lund-Mckay scores [16]. Riva et al. evaluated the patients before, during, and after RT (3 months). In agreement with Wang et al., a concurrent increase in mucosal and olfactory disorders was observed during RT. Furthermore, Riva et al. found that nasal symptoms and endoscopic findings peaked at the end of RT [15]. Further studies are necessary to identify where OAR should be set in order to reduce the incidence of sinonasal side effects.

The impact of CT on nasal side effects has been poorly analyzed. Only one study evaluated olfactory complaints during CT for HNC. The TDI score decreased during the second CT cycle, especially in older patients (>55 years), and reached almost its initial levels after 3 weeks of recovery time [59]. On the other hand, the impact of CT alone on sinonasal mucosa has never been investigated.

## 5. Conclusions

Sinonasal mucosa and smell disorders are a common post-treatment side effect of CT and/or RT in HNC patients. The incidence and severity of olfactory dysfunction and chronic rhinosinusitis were highest at the end of RT and at 3 months after treatment and decreased gradually with time. Smell acuity deterioration and chronic rhinosinusitis after RT seemed related to radiation dose on olfactory area and nasal cavities, but different degrees of recovery were observed. Therefore, it is important to establish the severity of chronic rhinosinusitis and olfactory dysfunction in order to find strategies to support patients and improve their quality of life. Further studies are necessary to better assess the role of medical and surgical treatments of sinonasal side effects of CT and/or RT for HNC. Finally, the role of CT should not be overlooked, and future studies are mandatory to assess its effect on nasal cavities and paranasal sinuses.

## Figures and Tables

**Figure 1 cancers-14-02324-f001:**
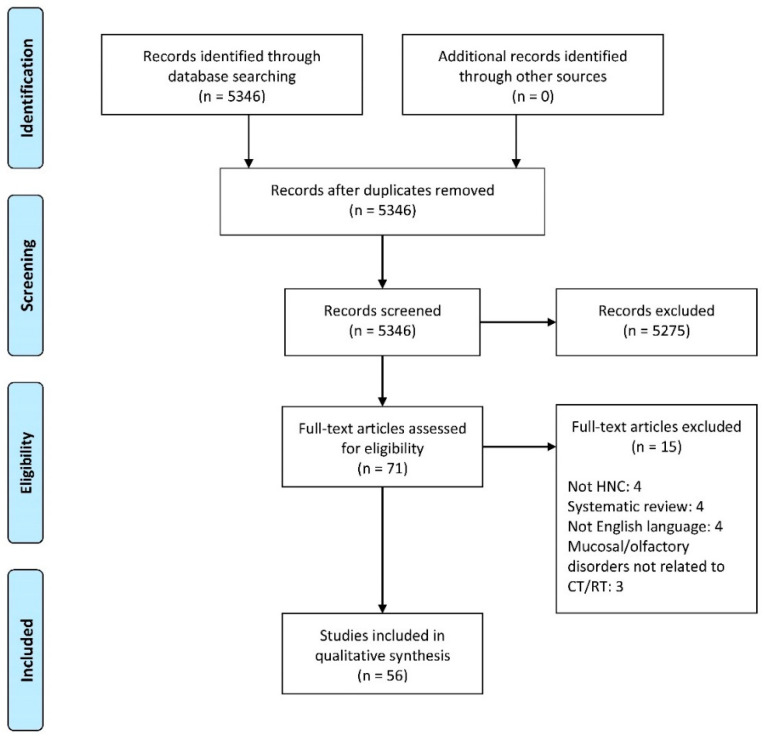
Flow diagram of the included studies.

**Table 1 cancers-14-02324-t001:** Sinonasal mucosa disorders: studies included in the review.

Author, Year, Country	Study Design	Number of Patients	Sex	Age, Mean and Range/ Standard Deviation (Years)	Tumor (Site and Stage)	Treatments	Measurements	Time of Assessment	Results
Stringer et al., 1995, USA [17]	Cross-sectional	Study group: *n* = 9 Control group: *n* = 9	Study group M: 6 (77%) F: 3 (33%) Control group: NR	Study group 80 (36–81) Control group: NR	Nasal vestibule or ala (*n* = 7), nasal cavity (*n* = 2), nasopharynx (*n* = 1) (stage NR)	RT (63.8–74.8 Gy) *Dose to nasal cavities:* NR	- MCC (saccharine test) - Subjective nasal symptoms	Before RT (subjective symptoms) and 20–117 months after RT (saccharine test and subjective symptoms)	- Reduced mucociliary clearance after RT - Higher prevalence of nasal congestion, drainage and facial pain after RT
Lou et al., 1999, Taiwan [18]	Cross-sectional	Study group: *n* = 10 (all with sinusitis) Control group: *n* = 6 (3 patients with sinusitis and 3 without)	Study group M: 7 (70%) F: 3 (30%) Control group: NR	Study group 45 (28–70) Control group: NR	Nasopharynx (stage NR)	RT (70–80 Gy) *Dose to nasal cavities:* Mean dose to infundibulum 21 Gy (17.5–25 Gy)	Biopsy of infundibulum mucosa (light and electron microscope views)	5.9 (0.8–23) years after RT	- Increased deposition of dense collagenous fibers in the lamina propria after RT - The epithelial cells transformed into a stratified arrangement and showed gradual reduction of cytoplasmic volume after RT - Areas of ciliary loss, intercellular and intracellular vacuolation, and ciliary dysmorphism after RT
Kamel et al., 2004, Egypt [19]	Retrospective	*n* = 32	M: 19 (59%) F: 13 (39%)	36 (7–65)	Nasopharynx (stage NR)	RT (doses NR) *Dose to nasal cavities:* NR	- MCC (saccharine test) - Nasal endoscopy - Computed Tomography scan (Lund-Mackay score)	Group I (*n* = 23): Saccharine test and nasal endoscopy before RT and at 2–6 weeks, 3 and 6 months, 1 and 2 years after RT; Computed Tomography scan 6–12 months after RT Group II (*n* = 9): 4–12 years after RT	- Increased saccharine delay time up to 6 months after RT, then it stabilized - Correlation between pre- and post-RT MCC delay time - Early edema and discharge (2–6 weeks after RT) and delayed crusting and adhesions (6 months after RT) - Maxillary sinus, anterior ethmoid sinus and ostiomeatal complex were the most affected regions - No correlation between RT dose and MCC delay time, endoscopic findings and Lund-Mckay score
Gupta et al., 2006, India [20]	Prospective	Study group: *n* = 50 Control group: *n* = 20	Study group M: 35 (70%) F: 15 (30%) Control group: NR	Study group 54.7 (35–78) Control group: NR	Larynx (*n* = 19), oropharynx (*n* = 15), oral cavity (*n* = 10), hypopharynx (*n* = 6) (stage I–IV)	RT (*n* = 14) (14–70 Gy) CT-RT (cisplatin, 5-fluorouracil, methotrexate) (*n* = 33) Surgery + RT (*n* = 3) *Dose to nasal cavities:* NR	MCC (saccharine test)	Before RT and 6 months after RT	- Higher saccharin perception time after RT compared to control group - Higher saccharin perception time in patients receiving RT dose >60 Gy - Higher saccharin perception time in patients receiving CT concurrent to RT
Hsin et al., 2007, Taiwan [21]	Cross-sectional	*n* = 20	M: 12 (60%) F: 8 (40%)	47.5 (22–69)	Nasopharynx (stage NR)	RT (70–76 Gy): - RT alone (*n* = 2) - CT-RT (cisplatin, 5-fuorouracil, *n* = 18) *Dose to nasal cavities:* NR	- Cultures (maxillary sinus specimens) - Radiographs	4.9 (0.5–21) years after RT	- 85% of culture were positive in acute maxillary sinusitis - Frequently identified aerobes and facultative anaerobes included alpha-hemolytic streptococcus (*n* = 8), *Staphylococcus aureus* (*n* = 5) and *Pseudomonas aeruginosa* (*n* = 3) - *Streptococcus pneumoniae*, *Haemophilus influenzae* and *Moraxella catarrhalis* were far less common
Huang et al., 2007, Taiwan [22]	Retrospective	*n* = 112	M: 77 (69%) F: 35 (31%)	47.9 (18.9–76.2)	Nasopharynx (stage I–IV)	RT (64–76 Gy): - external RT alone (*n* = 79) - external RT + brachytherapy boost (*n* = 33) Concurrent CT (cisplatin-based, *n* = 59) *Dose to nasal cavities:* NR	MRI scan (Lund-Mackay score)	Before RT, and at 3 months, 9 months, 2 years, 3 years, 4 years, and 5 years after RT	- Highest incidence and severity of SMD (sinus mucosa disease) at 3 months after RT (67.7% of cases) and decreased gradually with time - Most frequently affected sinuses were maxillary, anterior ethmoid, and posterior ethmoid sinuses - Advanced tumor stage and smoking habit were SMD predisposing factors (age, sex, RT dose, and nodal status were not) - No factors could predict SMD improvement after RT
Hu et al., 2008, Taiwan [23]	Prospective	Study group: *n* = 21 Control group: *n* = 10	Study group M: 13 (62%) F: 8 (28%) Control group: NR	Study group 49.5 (43–58) Control group: NR	Nasopharynx (stage NR)	RT (70–80 Gy) *Dose to nasal cavities:* NR	- MCC (saccharine test) - Mucosal specimens during FESS (electron microscope views)	Before and 1 year after FESS 2.1 (1.2–4.0) years between RT and FESS	- Choanal stenosis in 5 patients and nasal synechiae in 6 cases after RT - Decreased number of submucosal gland openings and ciliary area after RT - Regenerated cilia 1 year after FESS - No change in the number of goblet cells - Decreased saccharin transit time after RT
Liang et al., 2008, Taiwan [24]	Cross-sectional	Non irrigation group: *n* = 63 Irrigation group: *n* = 44	Non irrigation group M: 49 (78%) F: 14 (12%) Irrigation group M: 35 (79%) F: 9 (21%)	47.7 (17–81) Non irrigation group: 49.13 ± 1.81 Irrigation group: 45.61 ± 1.68	Nasopharynx (stage I–IV)	IMRT (56–76.8 Gy): - RT alone (*n* = 7) - CT-RT (*n* = 43) - induction CT +RT (*n* = 57) *Dose to nasal cavities:* NR	- Nasal endoscopy (Lund endoscopic staging system) - Computed Tomography scan (Lund-Mackay score) - Questionnaire on nasal symptoms	Before, at mid-course, and at the end of RT, 1, 2, 3, 6, and 12 months after RT (Computed tomography before RT and 3, 6 and 12 months after RT)	- Lower endoscopic and questionnaire scores in the irrigation from pre-RT to 6 months after RT - The between-group differences were most obvious at 2 and 3 months after RT - No differences in Lund-Mackay scores between the two groups from pre-RT to 6 months after RT
Deng et al., 2009, China [25]	Cross-sectional	*n* = 60 Post-RT CRS group: *n* = 30 CRS group: *n* = 30	Post-RT CRS group: M: 24 (80%) F: 6 (20%) CRS group: M: 23 (77%) F: 7 (23%)	Post-RT CRS group: 42.7 (23–70) CRS group: 33.8 (21–59)	Nasopharynx (stage NR)	RT (66–74 Gy) Adjuvant CT in 11 patients (5-fluorouracil and cisplatin) *Dose to nasal cavities:* NR	Cultures (maxillary sinus specimens)	2.92 (0.5–8.5) years after RT	- 73% of cultures after RT were positive - Isolated Gram-positive coccus rate in post-RT CRS patients was higher than in CRS patients (62.50% vs. 30.00%) - Isolated Gram-negative bacilli rate in post-RT CRS patients was lower than in CRS patients (31.25% vs. 70.00%) - The most common isolates in the post-RT CRS group were *Streptocuccus viridans*, *Staphylococcus aureus* and *Haemophilus influenzae*, while those in the CRS group were *Haemophilus influenzae*, *Pseudomonas aeruginosa* and *Staphylococcus aureus*
Lee et al., 2012, Taiwan [26]	Retrospective	*n* = 188	M: 132 (70%) F: 56 (30%)	49.49 (17–78)	Nasopharynx (stage I–IV)	2D-RT, 3D-CRT or IMRT (69.91±3.87 Gy): - RT alone (*n* = 100) - CT-RT (*n* = 88) *Dose to nasal cavities:* NR	- Computed Tomography scan - MRI scan - Nasal endoscopy	7.34 (3.30–26.54) years after RT	- CRS in 21.8% of patients - Choanal stenosis in 14.4% of cases
Xiang et al., 2013, China [27]	Retrospective	*n* = 40	M: 22 (55%) F: 18 (45%)	46 (23–65)	Nasopharynx (stage NR)	RT (68–72 Gy) *Dose to nasal cavities:* NR	- Nasal endoscopy - Computed Tomography/MRI scan - Subjective nasal symptoms (VAS)	3.4 (0–9) months after RT	- Nasal synechiae after RT were between the inferior turbinate and septum (100%), between the middle turbinate and septum (70%), between the inferior turbinate and nasal floor (50%), and between the middle turbinate and inferior turbinate (42.5%) - Patent nasal cavities in 95% of patients after surgery - Decreased post-operative VAS for nasal symptoms
Su et al., 2014, **China** [28]	Retrospective	*n* = 283	M: 215 (76%) F: 68 (24%)	48 (11–77)	Nasopharynx (stage II–IV)	IMRT (70.4–74.8 Gy): - RT alone (*n* = 29) - Induction CT + CT-RT (cisplatin and fluorouracil, *n* = 254) *Dose to nasal cavities:* NR	- Computed Tomography scan - MRI scan	Before and 1, 3, 6, 9, 12, and 18 months after RT	- Many patients with CRS before RT suffered aggravated symptoms after RT - 75.3% of patients without CRS before RT developed CRS after RT - Maxillary sinuses were the most common involved - Advanced T stage, invasion of the nasal cavity, and nasal irrigation (but not CT or RT dose) were positively associated with the incidence of sinusitis after RT - The incidence of sinusitis peaked at 6–9 months after RT and showed a trend toward stabilization after 1 year
Kılıç et al., 2014, Turkey [29]	Cross-sectional	*n* = 44	NPC group: M: 32 (45%) F: 12 (55%) Laryngeal cancer group: M: 22 (100%)	NPC group: 36 (18–63) Laryngeal cancer group: 56 (44–72)	Nasopharynx (*n* = 22), larynx (*n* = 22) (stage II–IV)	RT (70 Gy) Total laryncetomy and adjuvant RT in laryngeal cancer group *Dose to nasal cavities:* NR	MCC (saccharine test)	Before and 3 and 6 months after RT	- Higher MCC times 3 months after RT in both groups, in particular in NPC patients - MCC times decrease between 3 and 6 months after RT
Lou et al., 2014, China [30]	Cross-sectional	*n* = 1134 Group A (nasal irrigator): *n* = 378 Group B (homemade nasal irrigation connector combined with enemator): *n* = 378 Group C (nasal sprayer): *n* = 378	M: 826 (73%) F: 308 (27%) Group A: M: 268 (71%) F: 110 (29%) Group B: M: 273 (72%) F: 105 (28%) Group C: M: 285 (75%) F: 93 (25%)	48 (12–84) Group A: 43 (13–82) Group B: 51 (12–81) Group C: 49 (13–84)	Nasopharynx (stage I–IV)	RT (66–70 Gy): - 3D-CRT (*n* = 316) - IMRT (*n* = 818) CT in 972 patients (cisplatin and fluorouracil regimen or paclitaxel regimen) *Dose to nasal cavities:* NR	- MRI scan - SNOT-20	Before, and 6 months, 1, 2 and 3 years after RT	- Incidence of CRS was 42.6%, 56.3%, 86.1%, 75.8% and 69.7% at different time evaluations and was higher in group C after RT - Lower quality of life (SNOT-20) in group C compared to other groups after 1 year
Alon et al., 2014, Israel [31]	Retrospective	*n* = 62	M: 42 (68%) F: 20 (32%)	42 (11–74)	Nasopharynx (stage I–IV)	2D-RT/3D-CRT (*n* = 40 or) IMRT (*n* = 22) (66–72.4 Gy): - RT alone (*n* = 18) - CT-RT (cisplatin and 5-fluorouracil, *n* = 44) *Dose to nasal cavities:* NR	- Nasal endoscopy - SNOT-16	7 (3–16) years after RT	- CRS in 18% of patients, choanal stenosis in 15%, nasal synechiae in 7% of cases - CRS diagnosis was made 12 to 72 months after RT - No significant influence of RT delivery method for any type of complication - Choanal stenosis negatively affect quality of life (SNOT-16)
Hsin et al., 2015, Taiwan [32]	Retrospective	*n* = 102	M: 74 (73%) F: 28 (27%)	43.5 (19–74)	Nasopharynx (stage I–IV)	IMRT (68–81 Gy): - RT alone (*n* = 9) - CT-RT (*n* = 93) *Dose to nasal cavities:* NR	MRI scan (Lund-Mackay score)	Before and 5 years after RT	- CRS in 16.7% of patients 5 years after RT - Increase of Lund-Mackay score 5 years after RT - No significant association between the occurrence of middle ear toxicity and the Lund-Mackay score
Riva et al., 2015, Italy [33]	Cross-sectional	Study group: *n* = 30 Control group: *n* = 30	Study group M: 24 (80%) F: 6 (20%) Control group M: 20 (67%) F: 10 (33%)	Study group 53.53 (37–75) Control group 52.35 (42–76)	Nasopharynx (stage I–IV)	2D-RT (*n* = 5), 3D-CRT (*n* = 5), IMRT (*n* = 20) RT dose: 69.34 ± 1.17 Gy - Concurrent CT-RT (cisplatin-based, *n* = 4) - Concurrent CT-RT + adjuvant CT (*n* = 4) - Induction CT + concurrent CT-RT (*n* = 22) *Dose to nasal cavities:* NR	- Subjective nasal symptoms - Nasal endoscopy - Nasal cytology	59 (21–124) months after RT	- Higher percentage of rhinorrhea, nasal obstruction, mucosal hyperemia, and presence of nasopharyngeal secretions in the study group - Higher percentage of neutrophilic inflammation and squamous or mucous cell metaplasia the study group - No cytological atypia - No correlation between cytological changes and symptoms, endoscopic findings, age, smoking, tumor stage - No significant difference between different radiation techniques and radiation dose
Wang et al., 2015, Taiwan [16]	Prospective	*n* = 41	M: 31 (76%) F: 10 (24%)	45 (29–77)	Nasopharynx (stage I–IV)	IMRT (70–76.8 Gy): - RT alone (*n* = 2) - Concurrent CT-RT (*n* = 2) - induction CT + RT (*n* = 37) *Dose to nasal cavities:* NR	- Computed Tomography scan (Lund-Mackay score) - OI (UPSIT) - SNOT-22	Before and 12 months after RT	- Higher total and ethmoid Lund-Mackay score after RT - The decrease in UPSIT scores was moderately negatively correlated with the increase in total and ethmoid Lund-Mckay scores - The change in SNOT-22 scores was not significant, but the scores for item “loss of smell or taste” significantly increased after RT
Hsin et al., 2016, Taiwan [34]	Retrospective	*n* = 94	M: 67 (71%) F: 27 (29%)	42.7 (20–74)	Nasopharynx (stage I–IV)	IMRT (68–81 Gy): - RT alone (*n* = 10) - CT-RT (*n* = 84) *Dose to nasal cavities:* NR	MRI scan (Lund-Mackay score)	Before and 3 months, 1, 3, and 5 years after RT	- The rate and severity of sinus abnormalities were highest on the third month after RT - There was no significant increase in the incidence of abnormalities on the fifth year after RT compred to pre-treatment - The anterior ethmoid and maxillary sinuses were the most affected sinuses - No significant increase in the score for sinuses with preexisting abnormality on the third month after RT - Advanced T stage (but not RT dose) was positively associated with the incidence of sinus abnormality on the fifth year after RT
Feng et al., 2016, China [35]	Prospective	Intranasal steroid group (fluticasone propionate): *n* = 32 Nasal irrigation group: *n* = 31	Intranasal steroid group M: 13 (41%) F: 19 (59%) Nasal irrigation group M: 14 (45%) F: 17 (55%)	Intranasal steroid group: 38.86 ± 9.26 Nasal irrigation group: 39.36 ± 7.28	Nasopharynx (stage I–IV)	RT: - Intranasal steroid group: 67.57 ± 2.94 Gy - Nasal irrigation group: 66.28 ± 3.91 Gy *Dose to nasal cavities:* NR	- Subjective nasal symptoms (VAS) - Nasal endoscopy (Lund endoscopic staging system) - Computed Tomography scan (Lund-Mackay score) - SNOT-20	Before, and 3 and 6 months after RT	- Fewer nasal complaints (overall symptoms, blocked nose and headache), better quality of life and less severe endoscopic findings in steroid group at 3 and 6 months after RT - No difference in Lund-Mackay score between pre- and post-RT evaluations in both groups
Riva et al., 2017, Italy [36]	Cross-sectional	Study group: *n* = 25 Control group: *n* = 25	Study group M: 22 (88%) F: 3 (12%) Control group M: 19 (76%) F: 6 (24%)	Study group: 68.76 (50–83) Control group: 62.64 (48–76)	Larynx (stage II–IV)	Total laryngectomy: - without adjuvant RT (*n* = 15) - with adjuvant RT (*n* = 8) - with adjuvant CT-RT (*n* = 2) (RT doses NR) *Dose to nasal cavities:* NR	- Subjective nasal symptoms - Nasal endoscopy - Nasal cytology - Biopsy of inferior turbinate (light microscope views)	52 (26–97) months after treatment	- Mucous cell metaplasia in 20% of laryngectomized patients - Submucosal stromal fibrosis in all patients and submucosal inflammatory infiltrate in 1 case (9%) at histological examination - No correlation between cytological changes and symptoms, endoscopic findings (turbinate hypertrophy, mucosal hyperemia, nasal secretions), age, smoking, tumor stage, adjuvant RT
Kuhar et al., 2017, USA [37]	Retrospective	*n* = 114 CRSr: *n* = 15 CRSsNP: *n* = 43 CRSwNP: *n* = 56	CRSr: M: 6 (41%) F: 9 (59%) CRSsNP: M: 21 (49%) F: 22 (51%) CRSwNP: M: 25 (45%) F: 31 (55%)	CRSr: 58.1 (range NR) CRSsNP: 50.3 (range NR) CRSwNP: 50.9 (range NR)	Nasal cavity and paranasal sinuses (*n* = 12), nasopharynx (*n* = 1), skull base (*n* = 1), oral cavity (*n* = 1) (stage I–IV)	RT (30.75–129 Gy) *Dose to nasal cavities:* NR	- Computed Tomography scan (Lund-Mackay score) - Biopsy of sinonasal mucosa during FESS (light microscope views) - SNOT-22	5.73 ± 7.2 years after RT	- Increased squamous metaplasia and subepithelial edema in CRSr compared to CRSsNP - Fewer eosinophils per high-power field, less basement membrane thickening, and fewer eosinophil aggregates in CRSr compared to CRSwNP - Higher Lund-Mackay score in CRSr compared to CRSsNP - No SNOT-22 differences between CRSr and the other groups before FESS
Park et al., 2018, South Korea [14]	Retrospective	*n* = 186 RT group: *n* = 143 Non-RT group: *n* = 43	M: 162 (87%) F: 24 (13%) RT group: M: 124 (87%) F: 19 (13%) Non-RT group: M: 38 (88%) F: 5 (12%)	60.4 (47–83) RT group: 59.09 ± 11.64 Non-RT group: 64.70 ± 8.28	Nasopharynx (*n* = 24), oral cavity (*n* = 31), oropharynx (*n* = 46), hypopharynx (*n* = 23), larynx (*n* = 62) (stage I–IV)	RT group (60–70.4 Gy): - IMRT (*n* = 89) - 3D-CRT (*n* = 54) Concurrent CT (cisplatin and 5-fluorouracil) in 104 patients *Dose to nasal cavities:* NR	- Clinical examination - Computed Tomography/MRI scan	Every 3 months for 3 years after RT	- CRS 17.2% of patients (16.3% in non-RT group and 17.5% in RT group) - Maxillary sinus was most commonly involved (56.9%) - Higher bilateral CRS in non-RT group (85.7% vs. 60%) - No treatment difference (medical vs. surgical) between the two groups - Age, TNM stage, and underlying disease were not associated with the need for sinus surgery - Although RT itself was not associated with sinus surgery, concurrent CT was significantly associated with need for surgery
Shemesh et al., 2018, Israel [38]	Prospective	*n* = 9	M: 5 (55%) F: 4 (45%)	44.2 (15–74)	Nasopharynx (stage I–IV)	RT (66–70 Gy): - 2D-RT (*n* = 4) - IMRT (*n* = 5) *Dose to nasal cavities:* NR	- Computed Tomography scan (Lund-Mackay score) - SNOT-16	Before and 6 months after surgery	- 9 out 93 patients who underwent RT had surgery for sinonasal complications (6 with CRS, 4 with choanal stenosis, 2 with skull base osteoradionecrosis) - Post-operative reduction of Lund-Mackay score and improvement of quality of life (SNOT-16)
Hamilton et al., 2019, Canada [39]	Retrospective	*n* = 162	M: 80 (49%) F: 82 (51%)	31 (15–35)	Nasopharynx (*n* = 48), nasal cavity and paranasal sinuses (*n* = 9), oral cavity (*n* = 21), tonsil (*n* = 4), larynx (*n* = 11), salivary glands (*n* = 36), thyroid (*n* = 30), other (*n* = 3) (stage I–IV)	RT (40–70 Gy): - 3D-CRT (*n* = 152) - IMRT (*n* = 10) - brachytherapy (*n* = 11) Surgery in 96 patients CT (platinum-based) in 17 patients *Dose to nasal cavities:* NR	Clinical examination	Median follow-up: 6.4 years	- Nasal crusting in 16% of patients, epistaxis in 16%, and chronic sinusitis in 8% of cases
Stoddard et al., 2019, USA [40]	Retrospective	*n* = 22	M: 14 (67%) F: 8 (43%)	68.8 (50–88)	Nasal cavity, paranasal sinuses	RT (14.4–184.8 Gy): - RT alone (*n* = 8) - CT-RT (*n* = 13) *Dose to sinonasal cavities:* NR	Sinonasal swab specimens (routine culture and next-generation molecular gene pyrosequencing)	81.2 (1–156) weeks after RT	- *Staphylococcus aureus* was the most common organism identified by both culture and gene sequencing, followed by *Pseudomonas aeruginosa* - Gene sequencing identified pathogens differing from culture results in 50% of patients examined
Riva et al., 2019, Italy [15]	Prospective	*n* = 10	M: 10 (100%)	56.90 (39–72)	Nasopharynx (*n* = 3), oral cavity (*n* = 3), parotid gland (*n* = 3), primary unknown (*n* = 1) (stage I–IV)	Surgery (*n* = 8) Concurrent CT-RT (54–70 Gy) (*n* = 5) Induction CT + concurrent CT-RT (*n* = 1) *Dose to o nasal cavities:* - Mean dose (Dmean) to nasal cavities 13.59 ± 17.74 Gy - Near maximum dose (D2%) to nasal cavities 26.73 ± 31.80 Gy - Mean dose (Dmean) to inferior turbinate 18.90 ± 24.08 Gy - Near maximum dose (D2%) to inferior turbinate 26.46 ± 31.43Gy	- Nasal endoscopy - Nasal cytology - NOSE scale and subjective nasal symptoms - Mean dose (Dmean) and near maximum dose (D2%) to nasal cavities and inferior turbinates	Before (T0), at mid-course (T1), and at the end (T2) of RT, 1 and 3 months after RT (T3 and T4)	- Nasal symptoms and endoscopic findings peaked at the end of RT (T2) (rhinorrea in 70% of cases, crusting in 40%) - Nasal cytology showed that a radiation-induced rhinitis with neutrophils and sometimes bacteria occurred in 70% of cases and persisted after 1 month. Mucous cell metaplasia appeared in 10% of patients during RT and disappeared after 3 months. Squamous cell metaplasia was observed in 10% of cases only after the end of RT - Not significant increase of NOSE total score at T2 - Significant correlation between Dmean and D2% to inferior turbinates and neutrophilic rhinitis at T2, between D2% to inferior turbinates and mucous cell metaplasia at T2
Huang et al., 2019, Taiwan [41]	Retrospective	*n* = 230	M: 177 (77%) F: 53 (23%)	48.5 (18–80)	Nasopharynx (stage I–IV)	IMRT (54.45–70 Gy): - RT alone (*n* = 38) - CT-RT (cisplatin and 5-fluorouracil, *n* = 23) - Induction CT + RT/CT-RT (*n* = 169) *Dose to nasal cavities:* NR	Computed Tomography/MRI scan	Before and more than 6 months after RT	- Incidence of sinusitis was 54.3% before RT and 47% after RT - The presence of post-RT was a significant predictor for Disease-Free Survival, Freedom from local failure, and Freedom from distant failure, in addition to having high negative predictive value for local relapse (97.5%)
Lu et al., 2020, Taiwan [42]	Retrospective	*n* = 701	M: 625 (89%) F: 76 (11%) RT alone: M: 41 (82%) F: 9 (18%) Any-RT: M: 262 (89%) F: 33 (11%) No RT: M: 322 (90%) F: 37 (10%)	NR (>20)	Oral cavity (*n* = 479), nasopharynx (*n* = 97), hypopharynx (*n* = 59), oropharynx (*n* = 43), larynx (*n* = 32)	- RT alone (*n* = 50) - RT + any others treatments (*n* = 295) - No RT (*n* = 359) CT in 340 patients (RT doses NR) *Dose to nasal cavities:* NR	Clinical examination	More than 3 months after treatment	- Of the 701 patients, 7% experienced CRS within 5 after treatment - The RT-alone group, any-RT group, and no-RT group had 5-year incidence of CRS of 12%, 9.3%, and 4.5%, respectively - Patients in the RT-alone and any-RT groups exhibited an increased risk of CRS compared with patients in the no-RT group (hazard ratio: 6.76 and 2.91, respectively)
Yin et al., 2020, China [43]	Cross-sectional	*n* = 66	M: 46 (70%) F: 20 (30%)	38.76 (25–45)	Nasopharynx (stage I–IV)	IMRT (RT dose NR) CT (cisplatin and 5-fluorouracil) in 43 patients *Dose to nasal cavities:* 36.46 (23.14–56.38) Gy	- MCC (saccharine test) - Nasal endoscopy - MRI scan - SNOT-20	Before RT, and at the end of RT, and 3, 6, and 12 months after RT	- The threshold doses of IMRT ranged between 37 and 40 Gy - A low dose (< threshold dose) of IMRT was associated with higher mucocilia transport rate, better endoscopy test score, and improved SNOT-20 score - The patients who received IMRT at a dose less than the threshold had the least damaged nasal mucosa morphology, and functional impairment scores were highest 3 months after RT - Significant relationship between the turbinate thickness ratio and the radiation dose

Abbreviations: 2D-RT, Two-dimensional Radiotherapy; 3D-CRT, Three-Dimensional Conformal Radiotherapy; CRS, Chronic Rhinosinusitis; CRSr, radiation-induced Chronic Rhinosinusitis; CRSsNP, Chronic Rhinosinusitis without Nasal Polyps, CRSwNP, Chronic Rhinosinusitis with Nasal Polyps; CT, Chemotherapy; CT-RT, Chemoradiotherapy; F, Female; FESS, Functional Endoscopic Sinus Surgery; Gy, Gray; IMRT, Intensity Modulated Radiation Therapy; M, Male; MCC, Mucociliary Clearance; MRI, Magnetic Resonance Imaging; NOSE, Nasal Obstruction Symptom Evaluation; NPC, Nasopharyngeal carcinoma; NR, Not reported; OI, Odor identification; RT, Radiotherapy; SNOT, Sino-Nasal Outcome Test; UPSIT, University of Pennsylvania Smell Identification Test; VAS, Visual Analog Scale.

**Table 2 cancers-14-02324-t002:** Olfactory dysfunction: studies included in the review.

Author, Year, Country	Study Design	Number of Patients	Sex	Age, Mean and Range/ Standard Deviation (Years)	Tumor (Site and Stage)	Treatments	Measurements	Time of Assessment	Results
Ophir et al., 1988, Israel [44]	Prospective	*n* = 12	M: 9 (75%) F: 3 (25%)	54.8 (38–76)	Nasopharynx (*n* = 9), pituitary gland (*n* = 7) (stage NR)	2D-RT (66 Gy) No CT *Dose to olfactory area:* 25–28 Gy (nasopharyngeal carcinoma), 18–22 Gy (pituitary adenoma)	ODT (amyl acetate and eugenol)	Before RT, within a week after RT end, 1, 3 and 6 months later	- ODT increased for both compounds by the end of treatment - Worst olfactory ability: first week and 1 month after RT end - At 6 months after RT termination, ODT baseline levels were not yet recovered
Sagar et al., 1991, UK [45]	Retrospective	Study group: *n* = 25 Control group: *n* = 40	NR	NR	Nasopharynx, pituitary fossa, maxillary sinus (*n* = 25) (stage NR)	2D-RT (doses NR) No CT *Dose to olfactory area:* 50–75 Gy (study group)	Self-reported smell (ad hoc questionnaire)	During RT	- 15 patients (60%) reported an alteration of smell from the first treatment fraction diminishing toward RT end and ceasing after RT - Odor described as unpleasant and consistent with ozone
Hua et al., 1999, China [46]	Prospective	Study group (*n* = 49): - group 1 (awaiting RT): *n* = 24 - group 2 (after RT): *n* = 25 Control group: *n* = 36	Group 1 M: 16 (67%) F: 8 (33%) Group 2 M: 23 (92%) F: 2 (8%) Control group M: 26 (72%) F: 10 (28%)	Group 1 40.9 (27–59) Group 2 45.2 (28–60) Control group 43.6 (28–67)	Nasopharynx (T1-T3)	2D-RT (68–72 Gy) CT: NR *Dose to olfactory area:* NR	ODT (N-butyl alcohol), Odour Quality Discrimination test (5 odorants), Odour Recognition Memory Test, Odour-Visual Matching test, Odour-Tactile Matching test, OI (10 odorants), Odour Function test (edibility, function and identity)	Before RT (*n* = 24 NPC, group 1), after RT (*n* = 25 NPC, group 2)	NPC patients with RT had olfactory impairments including ODT, odour-tactile cross-modality matching, verbal identification of odours, recall and recognition of identity of odours
Ho et al., 2002, China [47]	Prospective	*n* = 48	M: 23 (48%) F: 25 (52%)	46 (22–71)	Nasopharynx (stage I–IV)	RT (*n* = 43) (doses NR) CT-RT (*n* = 15 *Dose to olfactory area:* NR	- ODT, OI, and OD (Sniffin’ Sticks) - Subjective hyposmia (VAS scale 0–100)	Before RT, end of RT, 3, 6 and 12 months after RT	- Deterioration of ODT and TDI score at 12 months - No changes in OD, OI and self-reported hyposmia (VAS) at 12 months
Hölscher et al., 2005, Germany [48]	Prospective	*n* = 44	M: 28 (64%) F: 16 (36%)	55 (11–81)	Maxillary sinus (*n* = 10), oropharynx (*n* = 10), oral cavity (*n* = 5), paranasal sinus (*n* = 5), nasopharynx (*n* = 6), hypopharynx (*n* = 2), nasal cavity (*n* = 1), brain (*n* = 1), skin (*n* = 1), unknown primary (*n* = 1), other (*n* = 2) (stage NR)	3D-CRT (30–76 Gy) (*n* = 30) CT-RT (*n* = 14) *Dose to olfactory area:* - OLF group (*n* = 22): >20 Gy (median 62.2 Gy, range 23.7–79.5 Gy) - Non-OLF group (*n* = 22): <12 Gy (median 5.9 Gy, range 2.9–11.1 Gy)	ODT, OI, and OD (Sniffin’ Sticks)	- Before and bi-weekly during RT for 6 weeks (*n* = 44) - Long term evaluation for 10 OLF patients (34 weeks after RT) and 15 non - OLF patients (39 weeks after RT)	- During RT: OD, but not ODT and OI, was significantly decreased 2–6 weeks after beginning of RT in the OLF group - Long term evaluation: lower OI, but not ODT and OD, in OLF vs. non-OLF group - Dose-effect relationship for OD (analyzing dose to olfactory epithelium) during RT, while after RT just a trend was found
Sandow et al., 2006, USA [49]	Prospective	Study group: *n* = 13 Control group: *n* = 5	Study group M: 10 (77%) F: 3 (23%) Control group M: 3 (60%) F: 2 (40%)	Study group 51.6 (40–75) Control group 47.9 (27–70)	Oropharynx (stage NR)	3D-CRT (63–76 Gy) CT-RT (cisplatin, *n* = 3) *Dose to olfactory area:* NR	OI (UPSIT)	Before RT, 1, and 12 months after RT	OI was unaffected by RT
Bindewald et al., 2007, Germany [50]	Cross-sectional	*n* = 205	M: 190 (93%) F: 15 (7%)	64 (32–84)	Larynx (stage I–IV)	Total laryngectomy (*n* = 20) Total laryngectomy + RT (*n* = 72) Partial laryngectomy (*n* = 77) Partial laryngectomy + RT (*n* = 36) (doses NR) CT: NR *Dose to olfactory area:* NR	Self-reported smell (EORTC QLQ-H&N35)	- 5.7 (0.11–16.58) years after total laryngectomy - 4.5 (0.19–15.14) years after partial laryngectomy	No differences in olfactory alterations between irradiated and non-irradiated patients
Rhemrev et al., 2007, The Netherlands [51]	Cross-sectional	*n* = 72	M: 44 (61%) F: 28 (39%)	57 (33–79)	Oral cavity, oropharynx (stage I–IV)	Surgery (*n* = 15) Surgery + RT (*n* = 57) (66–70 Gy) CT: NR *Dose to olfactory area:* NR	Self-reported smell (EORTC QLQ-H&N35)	43 (2–120) months after treatment	Higher olfactory alterations in irradiated patients
Brämerson et al., 2013, Sweden [52]	Prospective	*n* = 71	M: 51 (72%) F: 20 (28%)	60.9 (35–86)	Paranasal sinuses (*n* = 10), parotid gland/ear/facial skin (*n* = 8), oral cavity (*n* = 12), nasopharynx/larynx (*n* = 15), oropharynx (*n* = 26) (stage NR)	RT (*n* = 39) (doses NR) CT-RT (platinum compounds, pyrimidine compounds and taxanes, *n* = 32) *Dose to olfactory area:* - Low RT dose (*n* = 56): <10 Gy (mean 2.2 Gy) - High RT dose (*n* = 15): >10 Gy (mean 65.9 Gy)	- ODT, OI (SOIT) - Subjective hyposmia	Before RT and 20 (12–35) months after RT	- ODT and OI decreased after RT in both groups with a larger difference in the high-dose group - After therapy, 40% and 7% reported subjective olfactory decline in high and low RT dose groups, respectively - CT was not significantly different between high and low RT dose groups
Momeni et al., 2013, USA [53]	Cross-sectional	*n* = 21	M: 15 (71%) F: 6 (29%)	57.9 (24–87)	Oral cavity (*n* = 15), esophagus (*n* = 2), scalp (*n* = 2), pharynx (*n* = 1), paranasal sinus (*n* = 1) (stage NR)	Surgery (*n* = 8) Surgery + RT (*n* = 13) (doses NR) CT: NR *Dose to olfactory area:* NR	Self-reported smell (EORTC QLQ-H&N35)	24 (18–48) months after treatment	No differences in olfactory alterations between irradiated and non-irradiated patients
Oskam et al., 2013, The Netherlands [54]	Prospective	*n* = 80	M: 47 (59%) F: 33 (41%)	58 (23–74)	Oropharynx (*n* = 42), oral cavity (*n* = 38) (stage II–IV)	Surgery + RT (doses NR) CT: NR *Dose to olfactory area:* NR	Self-reported smell (EORTC QLQ-H&N35)	- Before treatment, 6 and 12 months after treatment - Long term evaluation: 9.2 (8–11) years (*n* = 27)	No statistically significant difference in taste/smell score among evaluations over time, but a deterioration was present after treatment
Jalali et al., 2014, Iran [55]	Prospective	*n* = 54	M: 26 (48%) F: 28 (52%)	49 (22–86)	Nasopharynx (*n* = 24), oropharynx (*n* = 6), paranasal sinus (*n* = 12), brain (*n* = 9), skin (*n* = 3) (stage NR)	RT (*n* = 30) CT-RT (*n* = 24) 12 patients with previous surgery Total RT dose: 50.1 Gy (range: 30–66 Gy) *Dose to olfactory area:* 334 μC (IQR 162–2068 μC)	ODT (N-butanol)	Before RT, during RT (2,4, 6 weeks), and after RT (3 and 6 months)	- ODT deteriorated during and after RT - No difference between ODT of patients according to radiation region or CT - The median cumulative local radiation for olfactory impairment (i.e., ODT ≤5) was 154 μC (IQR, 58–905 μC). - ODT significantly decreased 2–6 weeks after initiation of RT with cumulative dose of >135 μC
Veyseller et al., 2014, Turkey [56]	Cross-sectional	Study group: *n* = 24 Control group: *n* = 14	Study group M: 14 (56%) F: 10 (44%) Control group M: 5 (36%) F: 9 (64%)	Study group 48.7 ± 11.4 Control group 48.8 ± 7.0	Nasopharynx (stage I–IV)	CT-RT (68–72 Gy): - 2D-RT + cisplatin (*n* = 8) - 2D-RT + cisplatin and docetacel (*n* = 16) *Dose to olfactory area:* NR	- ODT and OI (CCCRC test) - Olfactory bulb volume (MRI scan)	66 (14–218) months after RT	- Lower ODT and OI in the NPC group compared to the control group - Lower mean olfactory bulb volume in the NPC compared to the control group - No significant differences in the olfactory bulb volume between different CT regimens
Riva et al., 2015, Italy [57]	Cross-sectional	Study group: *n* = 30 Control group: *n* = 30	Study group M: 24 (80%) F: 6 (20%) Control group M: 20 (67%) F: 10 (33%)	Study group 53.5 (37–75) Control group 52.3 (42–76)	Nasopharynx (stage I–IV)	CT-RT (cisplatin-based regimens): - 2D-RT/3D-CRT (*n* = 10): 70.2 Gy - IMRT (*n* = 20): 69–70 Gy *Dose to olfactory area:* NR	- ODT, OI, OD (Sniffin’sticks) - Subjective reduced or altered smell	59 (24–124) months after RT	- Higher percentage of reduced, but not altered, smell in study group - No differences for subjective hyposmia among radiation techniques - Higher ODT and TDI, but not OI and OD, in the control group compared to study group - No difference in ODT, OI and OD among radiation techniques
Landström et al., 2015, Sweden [58]	Prospective	*n* = 19	M: 12 (63%) F: 7 (37%)	56.6 (20–78)	Oral cavity (*n* = 18), oropharynx (*n* = 1) (stage I–IV)	ECT (bleomycin) (*n* = 6) ECT (bleomycin) + RT (57.8 Gy) (*n* = 13) *Dose to olfactory area:* NR	Self-reported smell (EORTC QLQ-H&N35)	Before treatment, and 12 months after treatment	No differences in problems with senses from baseline to 12 months after treatment
Haxel et al., 2015, Germany [59]	Prospective	*n* = 33	M: 25 (76%) F: 8 (24%)	61.6 (44–85)	Oropharynx (*n* = 20), larynx (*n* = 8), hypopharynx (*n* = 5) (stage NR)	CT (cisplatin, 5-fluorouracil and docetaxel) No RT *Dose to olfactory area:* NA	ODT, OI, OD (Sniffin’sticks)	Before and immediately after first, second and third CT cycle	- TDI score decrease during the second CT cycle was significant - Older patients (>55 years) were more susceptible to decreasing TDI score during first and second CT cycles - TDI score reached almost their initial levels after 3 weeks of recovery time
Wang et al., 2015, Taiwan [16]	Prospective	*n* = 41	M: 31 (76%) F: 10 (24%)	45 (29–77)	Nasopharynx (stage I–IV)	IMRT (70–76.8 Gy): - IMRT alone (*n* = 2) - Concurrent CT-RT (*n* = 2) - induction CT + IMRT (*n* = 37) *Dose to olfactory area*: NR	- OI (UPSIT) - Self-reported smell (SNOT-22)	Before and 12 months after RT	- Significant decrease in UPSIT score after RT - The change in SNOT-22 scores was not significant, but the scores for item “loss of smell or taste” significantly increased after RT - UPSIT scores negatively correlated with total and ethmoid Lund-Mckay scores
**Alvarez-Camacho et al.,** **2016, Canada** [60]	Prospective	*n* = 160	M: 126 (79%) F: 34 (21%)	58.9 ± 11.9	Pharynx (*n* = 88), larynx (*n* = 36), oral cavity (*n* = 18), salivary glands (*n* = 11), nasal cavity and paranasal sinuses (*n* = 6), soft tissue (*n* = 1) (stage I–IV)	Surgery (*n* = 7) Surgery + RT (60 Gy) (*n* = 59) Surgery + CT-RT (cisplatin or carboplatin) (*n* = 86) Surgery + RT + cetuximab (*n* = 8) *Dose to olfactory area:* NR	Self-reported smell (CCS)	Before treatment, end of treatment and at 2.5 months follow-up	- Smell perception was significantly impaired at the end of treatment, with a partial recovery at 2.5 months follow-up - CCS (including taste and smell) was a significant predictor of overall quality of life, social-emotional, physical and overall functions at UW-QoL
Galletti et al., 2016, Italy [61]	Cross-sectional	Study group: *n* = 9 Control group: *n* = 9	Study group M: 9 (100%) Control group M: 9 (100%)	Study group 55 ± 9.96 Control group 52.56 ± 8.56	Nasopharynx (stage III–IV)	Induction CT (cisplatin and fluorouracil) + concurrent CT-RT (cisplatin, 60–69 Gy) *Dose to olfactory area:* NR	- Olfactory event-related potential testing - Subjective hyposmia (VAS scale 0–10, and 6-item Hyposmia Rating Scale)	44.77 ± 25.93 months after treatment	- Significant differences in latency and amplitude of olfactory event-related potentials between patients and controls (worse in patients) - Significant negative correlation between olfactory event-related potentials and the 6-item Hyposmia Rating Scale - Significant positive correlation between olfactory event-related potentials and the VAS scale
Badr et al., 2017, USA [62]	Cross-sectional	*n* = 93	M: 73 (78%) F: 20 (22%)	61.5 (39–88)	Oral cavity (*n* = 26), oropharynx (*n* = 67) (stage I–IV)	RT (*n* = 3) (doses NR) CT-RT (*n* = 22) Surgery + RT (*n* = 32) Surgery + CT-RT (*n* = 36) *Dose to olfactory area:* NR	Self-reported smell (Vanderbilt Head and Neck Symptom Survey version 2.0)	Within 3 months of RT end (7% of participants), within 3–6 months (23%), within 6–9 months (24%) and within 9–12 months (46%)	- Younger patients (<60 years) reported more smell problems than older patients (>60 years) - Smell disorders were predictors of depression and anxiety
Riva et al., 2017, Italy [63]	Cross-sectional	Study group: *n* = 50 Control group: *n* = 50	Study group M: 43 (86%) F: 7 (14%) Control group M: 40 (80%) F: 10 (20%)	Study group 68.76 (50–83) Control group 67.54 (53–76)	Larynx (stage II–IV)	Total laryngectomy + RT (*n* = 16) Total laryngectomy + CT-RT (*n* = 4) *Dose to olfactory area:* NR	ODT, OI, OD (Sniffin’sticks)	61.96 (24–132) Months after treatment	- Significant decrease of ODT, OI, OD, and TDI score in the study group - No correlation between TDI score and RT, age, and follow-up time at multivariate analysis
Lilja et al., 2018, Finland [64]	Prospective	*n* = 44	M: 29 (66%) F: 15 (34%)	56.2 (38–80)	Oral cavity (*n* = 28), oropharynx (*n* = 13), hypopharynx (*n* = 3) (stage II–IV)	Surgery (*n* = 5) Surgery + RT (*n* = 35) Surgery + CT-RT (*n* = 4) *Dose to olfactory area:* NR	- ODT (phenylethyl methyl ethyl carbinol) - OD and OI (7 odours)	Before treatment, and 6 weeks, 3, 6 and 12 months after treatment	- No differences in ODT between pre- and post-treatment scores - Higher scores in the OD in the 6-week and 3-month tests compared with preoperative scores for the tumour side - Higher scores in the OI in all post-treatment tests compared with preoperative scores
Riva et al., 2019, Italy [15]	Prospective	*n* = 10	M: 10 (100%)	56.90 (39–72)	Nasopharynx (*n* = 3), oral cavity (*n* = 3), parotid gland (*n* = 3), primary unknown (*n* = 1) (stage I–IV)	Surgery (*n* = 8) Concurrent CT-RT (54–70 Gy) (*n* = 5) Induction CT + concurrent CT-RT (*n* = 1) *Dose to olfactory area:* - Mean dose (Dmean) to nasal cavities 13.59 ± 17.74 Gy - Near maximum dose (D2%) to nasal cavities 26.73 ± 31.80 Gy	- ODT, OI, OD (Sniffin’sticks) - NOSE scale and subjective reduced or altered smell	Before (T0), at mid-course (T1), and at the end (T2) of RT, 1 and 3 months after RT (T3 and T4)	- Although olfactory function remained within the normal range at the evaluated times, a significant decrease in ODT, OD and TDI score was observed during RT, which returned to baseline levels after RT - Not significant increase of NOSE total score at T2 - Near significant correlation between Dmean to nasal cavities and subjective hyposmia at T2 and between D2% to nasal cavities and dysosmia at T2
Epstein et al., 2020, USA [65]	Prospective	*n* = 10	M: 7 (70%) F: 3 (30%)	59.9 ± 7.0	Oropharynx (*n* = 9), oral cavity (*n* = 1) (stage I–III)	IMRT: - alone (*n* = 1) - CT-RT (platinum-based, *n* = 9) *Dose to olfactory area:* NR	OI (UPSIT)	4–6 weeks after starting of treatment (*n* = 6) and up to 2 years after treatment (*n* = 8)	Decreased OI in 3 patients (33%) during treatment with smell recovery after treatment
Tyler et al., 2020, USA [66]	Cross-sectional	*n* = 114	M: 68 (60%) F: 46 (40%)	55 (18–78)	Nasopharynx (*n* = 61), paranasal sinuses (*n* = 29), nasal cavity (*n* = 24) (stage I–IV)	IMRT (*n* = 110) (66.6 ± 5.1 Gy) 3D-CRT (*n* = 4) Surgery (*n* = 38) CT: - induction (*n* = 8) - induction + concurrent (*n* = 58) - concurrent (*n* = 12 - concurrent + adjuvant (*n* = 9) *Dose to olfactory area:* NR	Self-reported smell (EQ-5D VAS, MDASI-HN, ASBQ)	65 (12–154) months after treatment	- The most frequently reported high-severity items in ASBQ were difficulty with smell and nasal secretions - Negative correlation between MDASI-HN and ASBQ sum score - Positive correlation between EQ5D VAS score and ASBQ sum score
Gurushekar et al., 2020, India [67]	Prospective	*n* = 21	M: 16 (76%) F: 5 (24%)	42.62 (16–75)	Nasopharynx (*n* = 13) oropharynx (*n* = 4), oral cavity (*n* = 2), paranasal sinuses (*n* = 2) (stage NR)	CT-RT (*n* = 15) Surgery + RT (*n* = 4) RT (*n* = 2) (doses NR) *Dose to olfactory area:* NR	- ODT and OI (CCCRC test) - Self-reported smell (AHSP questionnaire) - Mucociliary clearance time (saccharine test)	Before RT, at mid-course of RT (*n* = 21), at the end of RT (*n* = 18), 3 months after RT (*n* = 13)	- OTD and OI showed significant reduction during RT with partial recovery at 3 months follow up - No significant deterioration of smell by AHSP, although overall QOL significantly deteriorated - Mucociliary clearance time prolonged in 72% of patients at the end of RT
Sharma et al., 2020, Denmark [68]	Cross-sectional	*n* = 27	M: 17 (63%) F: 10 (37%)	67 (47–83)	Nasal cavity (*n* = 19), paranasal sinuses (*n* = 8 (stage I–IV)	IMRT (60–68 Gy): - alone (*n* = 7) - Surgery + RT (*n* = 20) Concurrent CT (cisplatin, *n* = 6) *Dose to olfactory area:* NR	- OI (Brief Smell Identification Test) - Self-reported smell (SNOT-22)	6.4 (1.6–11.1) years after RT	- Impaired olfactory function in 63% of patients (76% with surgery and RT versus 17% in RT alone) - The risk of olfactory impairment increased with higher tumor stage
Alfaro et al., 2021, USA [69]	Cross-sectional	Study group: *n* = 40 Control group: *n* = 20	Study group M: 24 (40%) F: 16 (60%) Control group: M: 11 (55%) F: 9 (45%)	Study group 63 ± 12 Control group 58 ± 14	Oral cavity (*n* = 19), pharynx (*n* = 18), larynx (*n* = 3) (stage I–IV)	RT (alone or combined with surgery or CT, doses NR) (*n* = 40) Concurrent CT (*n* = 24) *Dose to olfactory area:* NR	- OI (UPSIT) - Smell intensity (general Labeled Magnitude Scale)	Between 6 months and 10 years after RT	- No differences in OI between study and control groups - Lower smell intensity when tasting caffeine solutions in the study group

Abbreviations: 2D-RT, Two-dimensional Radiotherapy; 3D-CRT, Three-Dimensional Conformal Radiotherapy; ASBQ, Anterior Skull Base Inevntory; AHSP, Appetite, Hunger and Sensory Perception; CCCRC, The Connecticut Chemosensory Clinical Research Center test; CCS, Chemosensory Complaint Score; CT, Chemotherapy; CTCAE, Common Terminology Criteria for Adverse Events; CT-RT, Chemoradiotherapy; ECT, Electrochemotherapy; EORTC QLQ-C30, European Organization for Research and Treatment of Cancer Quality of Life CoreQuestionnaire; EORTC QLQ-H&N35, EORTC Quality of Life Head and Neck Module 35; EQ-5D VAS, EuroQol Group-5 Dimension Visual Analogue Scale; F, Female; Gy, Gray; IMRT, Intensity Modulated Radiation Therapy; IQR, Inter-Quartile Range; M, Male; MDASI-HN, MD Anderson Symptom Inventory–Head and Neck; MRI, Magnetic Resonance Imaging; NA, Not Applicable; NOSE, Nasal Obstruction Symptom Evaluation; NPC, Nasopharyngeal carcinoma; NR, Not reported; OI, Odor identification; OD, Odor discrimination; ODT, Odor detection threshold; RT, Radiotherapy; SNOT, Sino-Nasal Outcome Test; SOIT, Scandinavian Odor Identification test; TDI, Threshold, discrimination and identification total score; UPSIT, University of Pennsylvania Smell Identification Test; UW-QoL, University of Washington Quality of Life Questionnaire; VAS, Visual Analog Scale.

## Data Availability

Not applicable.

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
