# Peer review of "Sinonasal Side Effects of Chemotherapy and/or Radiation Therapy for Head and Neck Cancer: A Literature Review"

_cancers, 2022, doi:10.3390/cancers14092324_

Round 1
Reviewer 1 Report
The authors have focused their attention on side effects of radio- and chemotherapy treated head and neck cancer. An attention of majority of researchers deals with an induction of secondary cancer, burning, digestive process irregularities etc. The choice of an induction of damage of sinonasal mucosa is quite proper, even when a question addresses more function than structure.
The following remarks should be taken under consideration.
- Post-therapeutic dysfunction of smell and taste could be an effect of COVID-19 infection. A small part of cited studies is derived from 2019-2022 that overlaps with COVID pandemia. It could modulate results.
- Concerning radiotherapy I would like to see an information on hearing loss following cis-Pt treatment that could give a better picture of various senses deficite in cancer patients.
- Are you sure that search initiation from the term “head and neck cancer” was a proper one? A disproportion between first (n= 5346) and final (n= 56) is drastic.
- Several literature references(e.g. 5, 6, 11, 27, 30, 41, 67, 69) seem to do not have sufficient bibliographic information.
Author Response
Thanks for your useful suggestions.
1. We specified the possibile role of COVID19 pandemia on results in the Discussion.
2. Information about hearing loss following cisplatin treatment was added in the Introduction.
3. We used a wider initial term to find all the studies that may be included in our literature review.
4. Literature references were corrected.
Reviewer 2 Report
The following remarks are minor comments:
In the results part :
- Table 2 (olfactory dysfunction) : the first column (author, year, country) is not fulfilled.
3.2. Sinonasal Mucosa Disorders
The authors should briefly describe some methods or techniques used to assess mucosal disorders, for example:
- Lund Endoscopic Staging System used in endoscopy ;
- Lund–Mackay staging systemused in imaging ;
- Saccharine test used to evaluate mucociliary clearance ;
“Lin et al. showed that the patients who received IMRT at a dose less than the threshold” : in reference 43, the first author is Yin.
3.3 Olfactory dysfunction
The authors should describe in more details the “olfactory event-related potential testing”.
Author Response
Thank you for your uselful suggestions.
- Tables were checked and corrected.
- Methods or techniques used to assess mucosal disorders has been described.
- Reference 43 was corrected.
- The “olfactory event-related potential testing” has been described.
Reviewer 3 Report
1. Where should the OAR be set to reduce the incidence of sinusitis and olfactory abnormalities?
2. What is the relationship between hypotension and abnormal orthostatic tolerance and chemoradiotherapy for head and neck cancer?
3. Is MRI more appropriate than CT for imaging evaluation?
Author Response
Thanks for your useful suggestions.
- A discussion about OAR set-up for reducing sinonasal complications was added in the Discussion section ("Further studies are necessary to identify where OAR should be set in order to reduce the incidence of sinonasal side effects").
- The relationship between hypotension and abnormal orthostatic tolerance and chemoradiotherapy for head and neck cancer was out of our aims.
- For rhinosinusitis CT scan is more appropriate than MRI scan. We specified this concept in the Results section.